# Implementation of prevention of mother-to-child transmission (PMTCT) in South Africa: outcomes from a population-based birth cohort study in Paarl, Western Cape

Jennifer Pellowski,[1] Catherine Wedderburn,[2] Jacob A M Stadler,[3,4] Whitney Barnett [3,4] Dan Stein,[5,6] Landon Myer,[7] Heather J Zar[3,4]

JP and CW are joint first authors.

For numbered affiliations see end of article.

**Correspondence to**
Dr Jennifer Pellowski;
jennifer_pellowski@brown.edu

## ABSTRACT

**Objectives** The coverage of prevention of mother-to-child transmission (PMTCT) services in South Africa is variable. Identifying gaps in the implementation of these services is necessary to isolate steps needed to further reduce paediatric infections and eliminate transmission.

**Setting** Two primary care clinics in Paarl, South Africa.

**Participants** 1225 pregnant women; inclusion criteria were 18 years or older, clinic attendance and remaining in area for at least 1 year.

**Methods** Data were collected through the Drakenstein Child Health Study, a population-based birth cohort in a periurban area of the Western Cape, South Africa. A combination of clinic records, hospital records, national database searches and maternal self-report were collected during the study.

**Results** Of the 1225 mothers enrolled in the cohort between 2012 and 2015, 260 (21%) were confirmed HIV infected antenatally and 1 mother tested positive in the postnatal period. Of those with documentation (n=250/260, 96%), the majority (99%) received antiretroviral prophylaxis or therapy (ART) before labour; however, there was a high rate of defaulting from ART noted during pregnancy (20%). All HIV-exposed infants with data received antiretroviral prophylaxis, 35% were exclusively breast fed until 6 weeks and 16% for 6 months. There were two cases of infant HIV infection (0.8%) who were initiated on ART but had complicated histories.

**Conclusion** Despite the low transmission rate in this cohort, reaching elimination will require further work, and this study illustrates several areas to improve implementation of PMTCT services and reduce paediatric infections including retesting at-risk HIV-negative mothers through the duration of breast feeding, infant HIV testing at any admission in addition to routine testing and improved counselling to prevent defaulting from treatment. Better data surveillance systems are essential for determining the implementation of PMTCT guidelines.

## INTRODUCTION

In the past decade, there has been widespread progress globally in the prevention of mother-to-child transmission (PMTCT) of

HIV and in 2014, the WHO launched the call for elimination of mother-to-child transmission (MTCT) of HIV.[1] Countries must meet specific criteria to achieve elimination status, including ≤50 new paediatric infections per 100 000 live births. For countries with high prevalence of antenatal HIV, these targets are very challenging and will only be achieved with extremely low transmission rates requiring almost total coverage of a comprehensive package of PMTCT interventions.

South Africa has the highest number of HIV-infected people in the world with prevalence rates of up to 40% among public antenatal clinic attendees.[2] In 2010, the Western Cape government rolled out guidelines, which, based on a pregnant woman's clinical and immunological status, provided antiretroviral therapy (ART) for life or zidovudine (AZT) starting at 14 weeks gestation (option

**BMJ**

A). In 2013, the Western Cape government rolled out option B+, which provides ART for life for all pregnant women regardless of CD4 T-cell count. The first-line regimen is triple therapy, comprising a non-nucleoside reverse-transcriptase inhibitor and two nucleoside reverse transcriptase inhibitors. Previous research, however, has shown that PMTCT service coverage in South Africa is variable, leading to missed opportunities for further reduction of transmission.[3]

A recent national evaluation[4] of the PMTCT programme showed a 6-week MTCT rate of 2.6% rising to 4.3% at 18 months post partum, indicating the need to further explore the actual implementation of PMTCT programmes in South Africa. Lessons from the implementation and uptake of PMTCT guidelines in the current era are necessary to inform how we may achieve very high coverage of a package of effective PMTCT interventions that will further reduce transmissions in high prevalence settings and achieve elimination. In this report, we sought to quantify the implementation of PMTCT guidelines using data from the Drakenstein Child Health Study,[5 6] a population-based birth cohort in a periurban area of the Western Cape, South Africa.

## METHODS

The Drakenstein Child Health Study recruited women during their second trimester of pregnancy (20–28 weeks gestation) from two community-based antenatal clinics in Paarl, South Africa.[5 6] Women were eligible for the study if they attended one of the two study clinics, planned to stay in the study area for at least 1 year and were 18 years or older. All participants provided informed consent, prior to participation in any study activities and were enrolled between 2012 and 2015.

### Data collection

Sociodemographic information was collected during the antenatal visit at 28–32 weeks gestation using structured interviews administered by trained study staff including: maternal age, education, marital status, employment and pregnancy planning.

HIV data were collected by triangulating clinic and hospital folder information and through self-report interviews completed with mothers during the antenatal period, at birth and post partum. Maternal HIV diagnosis was established at enrolment by self-report and confirmed during routine HIV testing of women in pregnancy as per the Western Cape PMTCT guidelines.[7 8] All HIV-infected mothers were enrolled into the Provincial PMTCT programme. As the recruitment for the cohort spanned changes in the Provincial PMTCT guidelines, mothers were initiated on ART per guidelines at the time: Option A before 2013, moving to option B+ in May 2013.

CD4 and viral load results were accessed from folder review and the online National Health Laboratory Service (NHLS) system. Where there was more than one result, the result closest to birth was used. ART was dichotomised

into first and second/third-line treatment, where the antiretroviral drug combinations relative to the guidelines at the time were used. Where available, data on regimen switching and treatment defaulting during pregnancy were also obtained. Defaulting was defined as an interruption (or discontinuation) of ART for at least a month during the index pregnancy.

Infant PMTCT data were obtained from infant clinic folder review and HIV test results from the NHLS system. Infant feeding method was obtained through self-report from mothers at 6 weeks, 10–14 weeks and 6 months post partum as previously reported.[9] Infant feeding was categorised into exclusive breast feeding, formula feeding or mixed breast feeding (breast feeding plus formula milk and/or solids). Ns, percentages, medians and interquartile ranges are reported for all variables of interest.

### Patient and public involvement

At the outset of the Drakenstein Child Health Study, close relationships were established with key stakeholders including Western Cape Government Health Department clinical and administrative staff and community members. Patients were not involved in the design or recruitment of the study, but at the 12 months postnatal study, visit participants were asked about their overall experiences with the study including experiences with study information, staff, procedures and how expectations about the study matched their experiences.[10] This information was used to reassess study protocols and to make alternations to improve study experience and participant satisfaction. Patients are not currently involved in dissemination plans, however, study findings are routinely fed back to the community and the Department of Health clinical and administrative staff.

## RESULTS

A total of 1225 women enrolled in the study. Overall, 260 (21%) pregnant women were confirmed to be HIV positive through antenatal hospital and clinic chart reviews (figure 1). At enrolment, women living with HIV were on average 29.1 years (SD=5.3), 25% were employed, 26% had completed secondary education and 59% of pregnancies were unplanned.

### Maternal interventions

Of the known women living with HIV with available data, the majority (182/255, 71%) knew their HIV status prior to the study pregnancy (table 1). One mother seroconverted in the postnatal period and thus, does not have antenatal PMTCT data. During the study pregnancy, most women had a CD4 T-cell count test (205/260; 79%), and the median CD4 count was 411 cells/mm$^3$ (IQR: 286–609) with 40% (82/205) having a CD4 of <350 cells/mm$^3$. Viral loads were available for 57% (148/260) of participants and most were undetectable (<40 copies/mL) (71%; 105/148). Prior to the 2013 guidelines, viral load testing

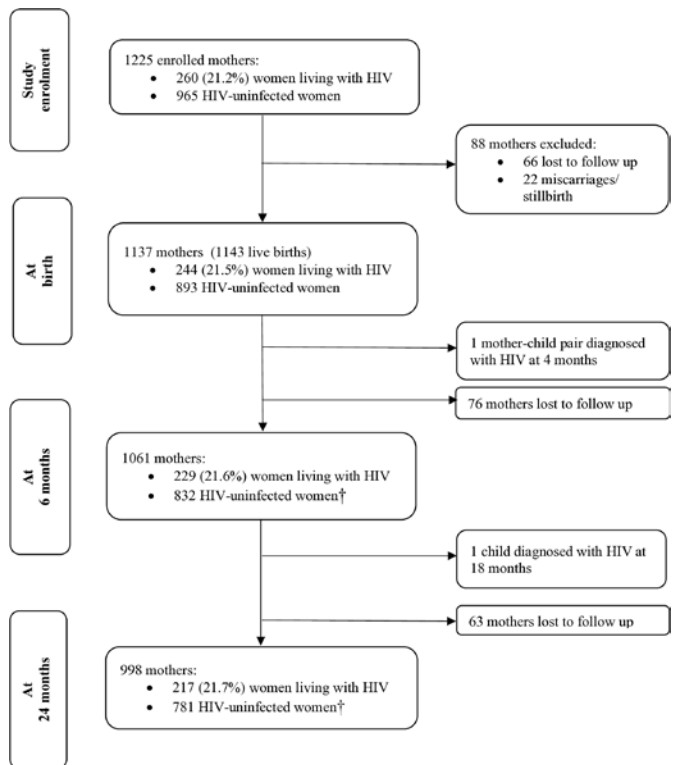

**Figure 1** Flow of participants through the Drakenstein Child Health Study. †Postnatal testing of women was not conducted through the cohort. Change in HIV status post partum was reported by women during detailed data collection about changes in their medical history at multiple time points.

was not routine during pregnancy and breast feeding for those on lifelong ART.

The majority of women received antiretroviral drugs during pregnancy (248/250; 99%; 10 missing data), 39 (16%) were on PMTCT antiretroviral prophylaxis as per option A (AZT), 195 (78%) were on first-line triple ART and 14 (6%) on second/third line. Two participants (0.8%) only received treatment during labour. There was evidence of defaulting from ART among 49 women (20%). Defaulters were not significantly different from non-defaulters by age, education, employment or unplanned pregnancy.

### Infant interventions

Of the 260 mothers, 5 had miscarriages/in utero deaths (including 1 set of twins), 11 were lost to follow-up before birth, and there were 248 recorded live births with 4 sets of twins (table 2). Of the live births, all infants with data were recorded as receiving prophylaxis; the majority of infants (211/242; 87%) received nevirapine. Six infants (2%) did not have data post partum, and receipt of ART prophylaxis is unknown. Only 35% (82/235) of mothers living with HIV reported exclusive breast feeding at 6 weeks post partum and 16% (37/235) for 6 months. A concerning 5.5% of mothers reported mixed feeding (breast feeding plus formula milk and/or/solids) at 6 weeks post partum, rising to 12% by 3 months.

| Table 1 | Maternal PMTCT results |
|---|---|
| **Variable (n, %)** | **Total (n=260)** |
| HIV diagnosis time point* | |
| Before pregnancy | 182 (71.4) |
| During pregnancy | 72 (28.2) |
| After birth | 1† (0.4) |
| CD4 (during pregnancy) | |
| CD4 cells/mm$^3$ (median) (IQR) | 411 (286–609) |
| ≥500 cells/mm$^3$ | 78 (30.0) |
| 350–500 cells/mm$^3$ | 45 (17.3) |
| <350 cells/mm$^3$ | 82 (31.5) |
| No CD4 reported during pregnancy | 55 (21.2) |
| Viral load (VL) (during pregnancy) | |
| Virally unsuppressed (≥1000 copies/mL) | 15 (5.8) |
| VL detectable (≥40–1000 copies/mL) | 28 (10.8) |
| VL lower than detectable limit (<40 copies/mL) | 105 (40.4) |
| No VL reported during pregnancy | 112 (43.1) |
| ART initiation* | |
| Before pregnancy | 98 (39.2) |
| During pregnancy | 150 (60.0) |
| Received ARVs during labour only | 2 (0.8) |
| ART regimen during pregnancy* | |
| PMTCT prophylaxis (AZT (zidovudine)) | 39 (15.6) |
| First-line ART (triple therapy) | 195 (78.0) |
| Second /third-line ART | 14 (5.6) |
| ARVs during labour only | 2 (0.8) |
| ART regimen switched during pregnancy* | 10 (4.0) |
| Evidence of defaulting any time during pregnancy* | 49 (19.6) |

Missing data: HIV diagnosis time point n=6; Antiretroviral therapy variables n=10.
*Percentages are calculated out of available data.
†This mother was diagnosed outside of the PMTCT pathway and as such she is not included in the other variables in this table.
ART, antiretroviral therapy; AZT, zidovudine;PMTCT, prevention of mother-to-child transmission; VL, viral load.

A total of 239 (96%) infants completed HIV PCR testing at 6–10 weeks post partum, which were all negative. At 9 months, 195 infants had HIV testing (79%); of these, 194 (99.5%) were negative and 1 (0.5%) was equivocal with a repeat test that was negative. As part of the larger cohort study, 18-month tests were followed up prospectively. Of the children who were still in the cohort at 18 months (n=220), 194 had a negative 18 month test, 22 had a negative test postcessation of breast feeding or were formula fed, 3 moved out of the area and 1 tested equivocal and then confirmed HIV positive. There was also testing that occurred outside of these standard tests when infant symptoms warranted testing. During these tests, one infant tested HIV positive.

**Table 2** Child PMTCT results

| Variable (n, %) | | | |
|---|---|---|---|
| Infant birth outcomes | **Total (n=265)** | | |
| Live births | 248 (93.6) | | |
| Stillbirths/miscarriages/in utero deaths | 6 (2.3) | | |
| Mothers lost to follow-up before/at birth | 11 (4.2) | | |
| Infant PMTCT outcomes | **Total (n=248)** | | |
| Infant prophylaxis* | | | |
| NVP prophylaxis | 211 (87.2) | | |
| NVP+AZT prophylaxis | 31 (12.8) | | |
| Feeding method at 6 weeks* | | | |
| Exclusively breast feeding | 82 (34.9) | | |
| Mixed feeding | 13 (5.5) | | |
| Formula feeding | 140 (59.6) | | |
| Infant testing | | | |
| 6–10 weeks post partum | | | |
| Positive | 0 (0) | | |
| Negative | 239 (96.4) | | |
| No test reported | 9 (3.6) | | |
| 9 months post partum | | | |
| Positive | 0 (0) | | |
| Negative | 194 (78.2) | | |
| Equivocal, repeat test negative | 1 (0.4) | | |
| No test reported | 53 (21.4) | | |
| HIV positive | | | |
| Yes | 2 (0.8)† | | |

*Percentages are calculated out of available data for live births in follow-up. Missing data: infant prophylaxis n=6; feeding method n=13.

†One infant born to newly diagnosed HIV-positive mother.

AZT, zidovudine; NVP, nevirapine.

### Mother-to-child transmission

Two infants in the cohort were confirmed HIV infected, translating to an MTCT rate of 0.8% (2 infants/261 mothers). In one case, the infant was born to a known HIV-infected mother with documented poor compliance to ART, poor maternal health and very poor social support structures. She booked at 18 weeks and had repeated viral load monitoring during pregnancy and post partum, which showed an unsuppressed viral load throughout pregnancy and breast feeding; she had already failed first-line therapy. The child was born at 39 weeks gestation. The mother opted for breast feeding, and the infant was covered with nevirapine prophylaxis; however, it is unclear if this was for the duration of exclusive breast feeding. Although the 6–10 weeks PCR and 9 months rapid test were negative, the infant had a positive rapid test at 18 months and was subsequently confirmed HIV positive by PCR and started on ART. However, both mother and child defaulted treatment and had to be restarted.

In the second case the mother tested HIV negative in the first trimester at booking (14 weeks) but subsequent testing was never done. The infant was born premature at 34 weeks gestation via vaginal delivery with a birth weight of 1.2 kg and was not managed according to PMTCT protocols, as the child was considered unexposed. Exclusive breast feeding was initiated, but the mother switched to formula feeding shortly after hospital discharge. Both mother and infant tested positive at 4 months of age when the infant was hospitalised with recurring pneumonia and was subsequently initiated on ART. This child was later diagnosed with tuberculosis and then developed drug-induced hepatitis and required specialist treatment.

### DISCUSSION

This study has shown a low rate of MTCT (0.8%) and reports on the entire PMTCT cascade with follow-up to 18 months. This may be due to the successful implementation of PMTCT guidelines in this periurban community, as demonstrated by the high rates of treatment in pregnant women and high rates of infant testing with low loss to follow-up compared with what has been previously reported.[11] However, one-fifth of women defaulted from antiretroviral drugs at some point during pregnancy, indicating a need for improved adherence counselling. Furthermore, nearly one-third of the sample had a CD4 T cell count of <350 with a large majority of women diagnosed prepregnancy, underscoring the importance of further focus on HIV care for women in general, outside of pregnancy.

Of the two HIV transmissions that occurred, both were identified after the 6–10 week infant PCR test. One transmission was to a known high-risk mother and the other to a mother diagnosed during the postnatal period. These findings highlight several gaps to address to achieve PMTCT goals. These include the importance of recognising high-risk mothers (eg, maternal poor adherence, first-line treatment failure, persistent unsuppressed viral loads) and providing increased support and careful follow-up of these mother–infant pairs throughout the duration of extended breast feeding.[12] Second, these findings increase awareness of infants and mothers who are diagnosed following symptomatic presentation and not via routine testing. These data reinforce the need to have a 'belts and braces' approach to infant testing, including HIV testing at any infant admission in addition to routine testing.[13] Third, it is important to retest HIV negative women during later pregnancy, during labour and during the breastfeeding period when women are at continued risk of transmitting acute infections to their infants. Fourth, pre-exposure prophylaxis, an effective form of HIV prevention, should be considered for at-risk HIV negative breastfeeding mothers in combination with standard sexual risk reduction methods. Finally, our sample had low rates of education and high rates of unplanned pregnancies, which are likely indicative of structural factors that may indirectly impede PMTCT

efforts, underscoring the need to focus on broader contextual contexts in addition to clinical efforts.

Data were obtained through a combination of clinic records, hospital records, national database searches and maternal self-report. Despite this exhaustive search, there were some missing data, indicating issues with PMTCT surveillance systems, a key foundational factor to achieving elimination of transmission. Although South Africa's NHLS system tracks CD4 and viral load testing results, integrated data systems that track adherence to all PMTCT guidelines and can account for movement between clinical facilities are lacking. Furthermore, systems such as NHLS are themselves an underused tool for tracking individual adherence to PMTCT guidelines. Training and resources are needed at the clinic level to identify and trace patients who may be missing these key PMTCT services.

There are some limitations to this study that should be considered. The women included in these analyses were enrolled in a birth cohort study and may have been more uniquely motivated to attend clinics and engage in care than women not enrolled in the study. Due to the eligibility criteria of the birth cohort study, participants had to present to the antenatal clinic in their second trimester. Thus, these results may indicate a 'best case' scenario for engagement in PMTCT interventions by excluding mothers living with HIV who are late presenters. Additionally, postnatal HIV testing of women who previously tested negative occurred outside of study procedures and was obtained through self-reports of significant maternal medical changes. Thus, the one mother who reported a postpartum diagnosis is potentially an underestimate for our cohort. Finally, this study, conducted in only two communities in the periurban area of Paarl, Western Cape, limiting our sample size of mothers living with HIV and the generalisability to other regions of the Western Cape or to other parts of South Africa. The Western Cape has some of the longest running PMTCT programme in South Africa and has some of the lowest MTCT rates in South Africa.[14] Our study highlights that even in areas where PMTCT programme are well functioning, there are still areas for improvement particularly in the context of elimination of MTCT.

## CONCLUSION

Although South Africa does not currently meet the criteria for elimination of MTCT, this study demonstrates that attaining extremely high levels of PMTCT coverage to further reduce transmission rates in high prevalence regions may be within reach. This may be achieved through retesting, breast feeding and ART adherence support, reinforced by improved data surveillance systems.

**Author affiliations**
[1]Department of Behavioral and Social Sciences and International Health Institute, Brown University School of Public Health, Providence, Rhode Island, USA
[2]Department of Clinical Research, London School of Hygiene and Tropical Medicine, London, UK
[3]Department of Paediatrics and Child Health, Red Cross War Memorial Children's Hospital, Rondebosch, South Africa
[4]Unit on Child and Adolescent Health, South African Medical Research Council, Tygerberg, South Africa
[5]Department of Psychiatry and Mental Health, University of Cape Town, Rondebosch, South Africa
[6]Unit on Risk and Resilience in Mental Disorders, South African Medical Research Council, Tygerberg, South Africa
[7]Division of Epidemiology and Biostatistics and Centre for Infectious Diseases Epidemiology and Research, University of Cape Town Faculty of Health Sciences, Observatory, South Africa

**Acknowledgements** We greatly thank the families and children who participated in this study. We would like to thank the study staff in Paarl, the study data team and laboratory teams, the clinical and administrative staff of the Western Cape Government Health Department at Paarl Hospital and at the clinics for support of the study. In particular, we thank Nienke Schalij, Julia Bondar and Mary Familusi for all their contribution to this work. We acknowledge the advice from members of the study International Advisory Board and thank our collaborators.

**Contributors** All listed authors meet criteria for authorship. Individuals who contributed to this manuscript but do not meet the criteria for authorship are listed in the acknowledgements section. JP and CW conceptualised the analysis, analysed the data and wrote the first draft of the paper. HJZ is principal investigator of the parent study. DS and LM are coinvestigators as contributed to study design and implementation. CW, JS and WB contributed to study implementation and data collection. All authors read and approved the final manuscript.

**Funding** The study was funded by the Bill and Melinda Gates Foundation [OPP 1017641]. Additional support was provided by the Medical Research Council of South Africa (HJZ and DJS), the SAMRC National Health Scholars programme (WB), the Wellcome Trust through a Research Training Fellowship [203525/Z/16/Z] (CJW) and the National Institute of Mental Health [K01 MH112443] (JAP).

**Competing interests** None declared.

**Patient consent for publication** Not required.

**Ethics approval** Ethics approval for this study was obtained through the Human Research Ethics Committee of the Faculty of Health Sciences, University of Cape Town, Stellenbosch University and the Western Cape Provincial Research committee.

**Provenance and peer review** Not commissioned; externally peer reviewed.

**Data availability statement** Data are available upon reasonable request.

**ORCID iD**
Whitney Barnett http://orcid.org/0000-0001-5082-7864

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
