## [Reviewer comments · BMJ Open]

ARTICLE DETAILS

TITLE (PROVISIONAL)	Implementation of prevention of mother-to-child transmission (PMTCT) in South Africa: Outcomes from a population-based birth cohort study in Paarl, Western Cape
AUTHORS	Pellowski, Jennifer; Wedderburn, Catherine; Stadler, Jacob; Barnett, Whitney; Stein, Dan; Myer, Landon; Zar, Heather

VERSION 1 – REVIEW

REVIEWER	KarlGunter Technau Empilweni Services and Research Unit, Department of Paediatrics and Child Health, Rahima Moosa Mother and Child Hospital, School of Clinical Medicine, Faculty of Health Sciences, University of the Witwatersrand, South Africa
REVIEW RETURNED	25-Aug-2019

GENERAL COMMENTS	The manuscript titled: "Implementation of prevention of mother-to-child transmission (PMTCT) in South Africa: Outcomes from a population-based birth cohort study in Paarl, Western Cape." presented by Pellowski et al is reviewed below. The authors present an interesting paper detailing the prevention of mother to child transmission (PMTCT) outcomes within a larger population based study in the Western Cape, South Africa. 1225 women were enrolled antenatally between 2012-2015. The authors describe HIV-prevalence, maternal diagnosis and treatment dynamics as well as infant diagnosis and prevention. The authors conclude that more work is required to reach elimination targets and provide suggestions as to how things may be improved. The manuscript is well written and presented. It is concise and I consider it a useful potential addition to the PMTCT literature. Regarding the reviewer checklist, overall the requirements are met but could be further improved taking into account points below. Regarding the checklists for cohort studies, the STROBE criteria seemed to have been adhered to. Possibly go through the list again once submitting the next version to ensure all is adhered to. There are a few points presented below that should be addressed/considered to possibly improve the manuscript: 1. General: Authors could consider the use of person first terminology, e.g. "woman living with HIV" instead of HIV positive woman.2. Abstract (applies equally to methods under "Data collection"). The concepts of defaulting, poor adherence and feeding methods play an important role in the manuscript and its conclusions. Could these perhaps be defined more clearly: e.g. how was defaulting and poor adherence defined. Were there specific time cutoffs of being missing from clinic follow-up, was it a more subjective
---

	measure. While these are not easy to measure it would help to clarify to the reader how the authors went about it. This would maybe also apply to further qualifying a statement like “compliance was questionable” for the first infected infant that is described. Similarly how was feeding defined and what qualified as “mixed”. 3. Good intro. 4. Good methods, but note 2. Above 5. Results (has implication for discussion): The authors detail 26% of women having completed secondary education and 59% of pregnancies being unplanned. For me this presents (possibly with other data not reflected but maybe available) to comment on the socio-economic situation as well as broader structural factors that are underlying. These factors may play a significant role in impeding progress towards EMTCT. This may warrant mention in the discussion, possibly mention of the SDGs. 6. Similar to Point above, 40% of women living with HIV had CD4<350, this suggests that there is still significant prevalence of immunocompromised in this largely pre-pregnancy diagnosed cohort. Improving on general female HIV care pre and during/post pregnancy may also impact MTCT rates, possibly more even than care during and after pregnancy. Maybe worth commenting on in discussion. 7. The authors mention the 260 women diagnosed prior to delivery. The abstract and result suggest that 260 women were diagnosed prior to delivery and 1 diagnosed after delivery. Am I reading this right, i.e. 261 total diagnoses? In this context it is important to reflect further on the one additional postnatal diagnosis. What was the denominator for this out of the 1225-260=965 women not diagnosed as positive. Out of the 965, what number of women had documented later testing, giving rise to the one additional case. The completeness of later maternal postnatal testing may hold an important key to there being a potential underestimate or not should other maternal postnatal infections have occurred that were not detected. It would also help in this context to state how many women had a 12 month follow-up. This will help to contextualize the 2/261 transmissions. Depending on the availability and completeness of maternal postnatal testing in the 1225 this may actually be a strength or alternatively a weakness of the study. 8. Results Page 7 lines 9-10: 5.5% mixed feeding at 6 weeks, 12% by 3 months. Is it possible to give the context of maternal VL in these mixed fed infant/mother pairs? With the growing emphasis on breastfeeding and maternal ART as the main protection for infants during breastfeeding, this information would be very useful. 9. Result page 7 Lines 15/16, 1 equivocal with repeat test negative. For this infant it would be very useful to know whether the equivocal test was during or after breastfeeding and therefore possibly during or after breast milk ART exposure, which potentially may have affected test sensitivity if the baby was infected. Was the repeat negative done after cessation of breastfeeding/maternal ART exposure? 10. Discussion Page 8 second sentence of second paragraph of discussion: “result of maternal seroconversion during pregnancy while breastfeeding”. This conclusion together with the concomitant results section detailing the scenario seems like an inaccurate interpretation. The mother was tested negative at first visit (14 weeks) and the next test, which showed her to be infected, was at 4 months postnatal. She could therefore have been infected from anytime between 12 weeks antenatal to postnatal. Given that the baby presented ill at four months, there is
--	---

	a high chance that this could even have been an in utero infection simply not detected due to incomplete maternal testing. The conclusion that it was a postnatal transmission may therefore be wrong and too specific, yes it could have been, but equally could have been an in utero infection or intra partum. This also affects the discussion later: more emphasis (not taking away from postnatal maternal testing) should be placed on later in-pregnancy and at delivery testing.
--	--

REVIEWER	Debra Jackson UNICEF
REVIEW RETURNED	28-Aug-2019

GENERAL COMMENTS	This is an excellent paper. I would like to see additions to the discussion and limitations. The sample size is also relatively small - this could be added to the sentence citing that the study was only in 2 sites in the limitations paragraph. More importantly, in many studies, the Western Cape, and Paarl in particular, have been shown consistently to have the best functioning PMTCT programmes and lowest transmission rates. While the authors acknowledge that these results may not be generalisable to the rest of the Western Cape or the country, I think from the literature the expected differences of this site versus the rest of South Africa could be discussed in a bit more detail. This would deepen the interpretation and recommendations beyond the current study.
---

VERSION 1 – AUTHOR RESPONSE

Reviewer: 1:

Please leave your comments for the authors below

The manuscript titled:” Implementation of prevention of mother-to-child transmission (PMTCT) in South Africa: Outcomes from a population-based birth cohort study in Paarl, Western Cape.” presented by Pellowski et al is reviewed below.

The authors present an interesting paper detailing the prevention of mother to child transmission (PMTCT) outcomes within a larger population based study in the Western Cape, South Africa. 1225 women were enrolled antenatally between 2012-2015. The authors describe HIV-prevalence, maternal diagnosis and treatment dynamics as well as infant diagnosis and prevention. The authors conclude that more work is required to reach elimination targets and provide suggestions as to how things may be improved.

The manuscript is well written and presented. It is concise and I consider it a useful potential addition to the PMTCT literature. Regarding the reviewer checklist, overall the requirements are met but could be further improved taking into account points below. Regarding the checklists for cohort studies, the STROBE criteria seemed to have been adhered to. Possibly go through the list again once submitting the next version to ensure all is adhered to.

Response: We now provide the STROBE checklist as a supplemental document.

There are a few points presented below that should be addressed/considered to possibly improve the manuscript:

1. General: Authors could consider the use of person first terminology, e.g. “woman living with HIV” instead of HIV positive woman.

Response: We have edited the manuscript to use person first language.

2. Abstract (applies equally to methods under “Data collection”). The concepts of defaulting, poor adherence and feeding methods play an important role in the manuscript and its conclusions. Could these perhaps be defined more clearly: e.g. how was defaulting and poor adherence defined. Were there specific time cutoffs of being missing from clinic follow-up, was it a more subjective measure. While these are not easy to measure it would help to clarify to the reader how the authors went about it. This would maybe also apply to further qualifying a statement like “compliance was questionable” for the first infected infant that is described. Similarly how was feeding defined and what qualified as “mixed”.

Response: We have edited the abstract and the related sections of the methods section to more clearly define defaulting/poor adherence/compliance and infant feeding. For the purposes of this analysis, both defaulting and adherence are in relation to maternal treatment. Defaulting treatment was defined as an interruption (or discontinuation) of antiretroviral treatment for at least a month during the index pregnancy. Data on treatment initiation date and any interruptions were collected from maternal folder reviews and documented along with restart dates. Where we refer to adherence counselling we are referring to preventing treatment discontinuation.

A detailed folder review of the two mother-child pairs with infected infants. The statement ‘compliance was questionable’ was reported from the infant medical notes, however, the details of how this statement was arrived at are unclear and thus we have opted to omit this from the manuscript.

Feeding methods were documented at 6-10 weeks, 14 weeks, 6 months and initiation of breastfeeding (and duration), formula milk and solids were all captured at each visit. Infant feeding was then categorised into exclusive breastfeeding, formula feeding or mixed breastfeeding (breastfeeding plus formula milk or solids) at each time-point.

3. Good intro.

4. Good methods, but note 2. Above

5. Results (has implication for discussion): The authors detail 26% of women having completed secondary education and 59% of pregnancies being unplanned. For me this presents (possibly with other data not reflected but maybe available) to comment on the socio-economic situation as well as broader structural factors that are underlying. These factors may play a significant role in impeding progress towards EMTCT. This may warrant mention in the discussion, possibly mention of the SDGs. Response: We agree with the reviewer’s point about education and the implications of broader structural factors. To address this we have added more information about this point in the discussion.

6. Similar to Point above, 40% of women living with HIV had CD4<350, this suggests that there is still significant prevalence of immunocompromised in this largely pre-pregnancy diagnosed cohort. Improving on general female HIV care pre and during/post pregnancy may also impact MTCT rates, possibly more even than care during and after pregnancy. Maybe worth commenting on in discussion. Response: This is an excellent point and we have added further discussion around CD4 and the immune status among our participant in relation to HIV care for women in general to the discussion.

7. The authors mention the 260 women diagnosed prior to delivery. The abstract and result suggest that 260 women were diagnosed prior to delivery and 1 diagnosed after delivery. Am I reading this right, i.e. 261 total diagnoses? In this context it is important to reflect further on the one additional postnatal diagnosis. What was the denominator for this out of the 1225-260=965 women not diagnosed as positive. Out of the 965, what number of women had documented later testing, giving rise to the one additional case. The completeness of later maternal postnatal testing may hold an important key to there being a potential underestimate or not should other maternal postnatal infections have occurred that were not detected. It would also help in this context to state how many

women had a 12 month follow-up. This will help to contextualize the 2/261 transmissions. Depending on the availability and completeness of maternal postnatal testing in the 1225 this may actually be a strength or alternatively a weakness of the study.

Response: The reviewer is correct in that there were 261 total diagnoses (260 diagnosed prior to delivery and 1 diagnosed after delivery). Of the 965 women not diagnosed as positive it is unclear how many received HIV testing in the postnatal period. The guidelines advised retesting every 12 weeks while breastfeeding and this was performed by clinical staff during routine care. The study conducted interviews and clinical reviews of mothers and children regularly over the postnatal period (at birth, 6 weeks, 14 weeks, 6 months, and then every 6 months. During these sessions participants were asked in detail about any significant medical changes (including HIV diagnosis and new medications) and so we hope to have captured any new diagnoses. However, the lack of consistent postnatal testing is a clear limitation of this work and part of the national guidelines more generally. We have added this as a limitation. Additionally, we include a new study flow chart documenting the numbers of women living with HIV through the study for clarity.

8. Results Page 7 lines 9-10: 5.5% mixed feeding at 6 weeks, 12% by 3 months. Is it possible to give the context of maternal VL in these mixed fed infant/mother pairs? With the growing emphasis on breastfeeding and maternal ART as the main protection for infants during breastfeeding, this information would be very useful.

Response: We agree that this is an important piece of information in the discussion of breast and mixed feeding. Of those mother/infant pairs where there was mixed feeding at 6 weeks, 8/13 had a VL antenatally, and 3 were detectable (>40 copies). However, we only have 1 postnatal viral load available (which was undetectable). Of those mother/infant pairs where there was mixed feeding at 3 months, 19/28 had a VL antenatally, and 5 were detectable (>40 copies). However, there were only 5 viral loads available in the early postnatal period and all were below the detectable limit. This will be an interesting avenue to explore further.

9. Result page 7 Lines 15/16, 1 equivocal with repeat test negative. For this infant it would be very useful to know whether the equivocal test was during or after breastfeeding and therefore possibly during or after breast milk ART exposure, which potentially may have affected test sensitivity if the baby was infected. Was the repeat negative done after cessation of breastfeeding/maternal ART exposure?

Response: This child was documented to be exclusively breastfed for the first 6 months of life and then formula and solids were introduced. The child received extended NVP prophylaxis for 36 weeks. The equivocal test was at nine months of age but this child had a negative rapid test at 18 months after cessation of breastfeeding and maternal ART exposure. Of interest, the child who was diagnosed with HIV infection at 18 months was also breastfed and had extended NVP prophylaxis and had an equivocal test result initially aged 18 months with a repeat positive test. We mention this in page 7, line 19. In this case it is possible the breast milk ART exposure may have affected test sensitivity as this child was found to be infected on retesting.

10. Discussion Page 8 second sentence of second paragraph of discussion: "result of maternal seroconversion during pregnancy while breastfeeding". This conclusion together with the concomitant results section detailing the scenario seems like an inaccurate interpretation. The mother was tested negative at first visit (14 weeks) and the next test, which showed her to be infected, was at 4 months postnatal. She could therefore have been infected from anytime between 12 weeks antenatal to postnatal. Given that the baby presented ill at four months, there is a high chance that this could even have been an in utero infection simply not detected due to incomplete maternal testing. The conclusion that it was a postnatal transmission may therefore be wrong and too specific, yes it could have been, but equally could have been an in utero infection or intra partum. This also affects the discussion later: more emphasis (not taking away from postnatal maternal testing) should be placed on later in-pregnancy and at delivery testing.

Response: The reviewer's point is fair and we have altered the language to be less specific about when the transmission was likely to have occurred. We have also modified our language in the discussion section to encompass testing later in pregnancy, at delivery, and postnatally.

Reviewer: 2

Please leave your comments for the authors below

This is an excellent paper. I would like to see additions to the discussion and limitations. The sample size is also relatively small - this could be added to the sentence citing that the study was only in 2 sites in the limitations paragraph. More importantly, in many studies, the Western Cape, and Paarl in particular, have been shown consistently to have the best functioning PMTCT programmes and lowest transmission rates. While the authors acknowledge that these results may not be generalisable to the rest of the Western Cape or the country, I think from the literature the expected differences of this site versus the rest of South Africa could be discussed in a bit more detail. This would deepen the interpretation and recommendations beyond the current study.

Response: We have added the sample size limitation to the discussion. We have also added more discussion about the differences between Western Cape/Paarl and the rest of the country. We feel that a key take away from this work is that even in areas where PMTCT programmes seem to be functioning the best, there are still areas for improvement particularly in the context of eMTCT.

VERSION 2 – REVIEW

REVIEWER	Karl-Gunter Technau University of the Witwatersrand, South Africa
REVIEW RETURNED	26-Oct-2019
GENERAL COMMENTS	The responses to the reviews for well done.
REVIEWER	Debra J Jackson UNICEF, USA
REVIEW RETURNED	17-Oct-2019
GENERAL COMMENTS	Excellent Revision, I think the minor addition makes your conclusions even stronger.